# Robust Superhydrophobic Brass Mesh with Electrodeposited Hydroxyapatite Coating for Versatile Applications

**DOI:** 10.3390/molecules27175624

**Published:** 2022-08-31

**Authors:** Yu-Ping Zhang, Shi-Ming Zhang, Peng-Fei Liu, De-Liang Chen, Yuan Chen, Meng-Jun Chen, Chang-Hua Zhao

**Affiliations:** 1College of Chemistry and Materials Engineering, Hunan University of Arts and Science, Changde 415000, China; 2Henan Institute of Science and Technology, Xinxiang 453000, China; 3Changde Zhengyang Biotechnology Co., Ltd., Changde 415000, China; 4College of Chemistry, Zhengzhou University, Zhengzhou 450001, China

**Keywords:** hydroxyapatite, superhydrophobic surface, anti-corrosion, critical-micelle-concentration

## Abstract

A robust superhydrophobic brass mesh was fabricated based on a low-energy surface and a roughness on the nano/micro-meter scale. It was carried out by the forming of hydroxyapatite (HP) coatings on its surface through a constant current electro-deposition process, followed by immersion in fluoroalkylsilane solution. Surface morphology, composition and wetting behavior were investigated by field-emission scanning electron microscopy (SEM), X-ray diffraction (XRD), X-ray photoelectron spectroscopy (XPS), high speed camera, and contact angle goniometer. Under optimal conditions, the resulting brass mesh exhibited superhydrophobicity, excellent anti-corrosion (η = 91.2%), and anti-scaling properties. While the surfactant liquid droplets of tetradecyl trimethyl ammonium bromide (TTAB) with different concentration were dropped on the superhydrophobic surface, maximum droplet rebounding heights and different contact angles (CAs) were observed and measured from side-view imaging. The plots of surfactant-concentration−maximum bounding height/CA were constructed to determine its critical-micelle-concentration (CMC) value. Close CMC results of 1.91 and 2.32 mM based on the determination of maximum rebounding height and CAs were obtained. Compared with its theoretical value of 2.1 mM, the relative errors are 9% and 10%, respectively. This indicated that the novel application based on the maximum rebounding height could be an alternative approach for the CMC determination of other surfactants.

## 1. Introduction

Nature is colorful, and many parts of animals and plants, such as a lotus leaf [1], red rose petal [2], peanut leave [3], butterfly wing [4], water strider leg [5], fish scale [6] and etc. are known for their superhydrophobic properties. Inspired by their excellent water repellency, many artificial superhydrophobic surfaces had been fabricated for versatile applications, such as self-cleaning, anticorrosion, anti-icing, anti-scaling, oil-water separation, etc. [7,8]. The wetting property at solid and liquid interfaces is intimately related to surface chemistry and roughness, and the coating is an essential step in regulating the surface properties of materials. Superhydrophobic coatings with static water contact angles (WCAs) greater than 150° and water slide angles (WSAs) below 10° have been developed, based on low-energy surfaces and roughness on the nano- and micrometer scales. In general, the WCAs are used to evaluate the static wettability of a solid surface, while WSAs or advancing/receding angles are usually applied to characterize the dynamic wettability of a solid surface. 

Hydroxyapatite (Ca_10_(PO_4_)_6_(OH)_2_, HA), as an important coating material, was generally used to improve the bioactivity and biocompatibility of titanium, magnesium, and their alloys [9]. The HA coating can be fabricated using plasma spraying [10], electrostatic spray deposition [11], hydrothermal method [12], electrophoretic deposition [13], sol-gel process and dip coating [14,15]. Though plasma spraying technique is the only clinically accepted method to deposit HA, it has several shortages including the usage of extremely high processing temperatures, the creation of a non-homogenous coating, and poor adhesion due to the delamination of HA from the substrate. As for electro-deposition, the acquisition of HA coating on a porous substrate has abundant ion sources (e.g., calcium chloride and dipotassium phosphate are easily available) and the manufacturing is simple and environmentally friendly without pollution and avoid the usage of complex and expensive equipment [16]. During electro-deposition, the electrolysis of water provides the hydroxyl groups needed to form hydroxyapatite and a large amount of hydrogen bubbles. These bubbles adhere to the surface of the deposited substrate, seriously hindering the nucleation and further growth of HA crystals, resulting in the HA coating with a rough and irregular surface [13].

In this article, we have described the fabrication of a uniform HA coating on the surface of brass mesh by employing the electro-deposition method under magnetic stirring. The deposition time, the selection of electrolyte with different ion species, pH, deposition temperature and current density were effectively optimized. After post-modification for the as-prepared brass mesh with HA coating, the superhydrophobic mesh with a thin layer of low surface energy material was used for the investigation of versatile applications such as oil/water separation, anti-corrosion, and anti-scaling, especially for the CMC determination of TTAB surfactant. 

## 2. Experimental Section

### 2.1. Fabrication of the Superhydrophobic Brass Mesh

The brass mesh (20 mm × 40 mm, Kangwei Wire Mesh Co. Ltd., Hebei, China) was used as the anode, whereas a platinum mesh of the same size was used as the cathode. Both electrodes were placed in a mixed electrolyte solution and positioned face to-face 20 mm apart. Prior to electrodeposition, both electrodes were sonicated for 5 min by immersion in acetone, ethanol and deionized water, respectively. Additionally, the brass meshes were immersed into an acid solution of HCl (2%) for ultra-sonication about 15 min to remove the native oxide film on the surface. Electrochemical deposition was performed under magnetic stirring from 10 to 70 min at a current density of 6.5 to 15 mA/cm^2^ and at an electrolyte temperature of 25–75 °C [9]. The meshes were then ultrasonically triple-rinsed by water, and subsequently dried with a hair dryer. After electrodeposition, the prepared meshes were then immersed in a 1.0 wt% ethanol solution of 1H,1H,2H,2H-Perfluorooctyltriethoxysilane (FOTS) for 3 h and then heated at 100 °C for 1 h. Robust and uniform structures were formed by electrochemical deposition with an optimal constant current of 120 mA at 75 °C in a water bath. The mixed electrolytes were composed of 4 mM Ca(NO_3_)_2_, 62.5 mM KH_2_PO_4_, 100 mM KNO_3_ and 4 mM Na_3_C_6_H_5_O_7_ solution, and pH of the electrolyte is adjusted to 5.6 using 0.1 M nitric acid [17]. The instrument of electrochemical deposition and reactions occurring on the cathode and anode surfaces are schematically illustrated in Figure 1.

### 2.2. Characterization

The surface morphologies of the meshes were observed under a scanning electron microscope (SEM, FEI Company, Hillsboro, OR, USA). The surface elements and composition of different substrates were measured by X-ray photoelectron spectroscopy (XPS, Thermo Fisher, Waltham, MA, USA). The crystal structures of the specimens were examined using an X-ray powder diffractometer (Bruker D8-Advance, Germany). The surface topography and nanoscale asperities were measured using a three-dimensional (3D) surface profilometer. The roughness of the surfaces was measured using an optical profilometer (GTK-16-0300, Bruker Scientific Company, Waltham, MA, USA). The contact and rolling angles were measured at ambient temperature via the pendent drop method using a contact angle goniometer (TST-200, Shen Zhen Testing Equipment Co. LTD, Shen Zhen, China). Water droplets (10 μL) were carefully dropped onto the surface of meshes, and the average of five measurements obtained at different positions in the meshes was adopted as the final contact angle. 

## 3. Results and Discussion

As shown in Figure 1, the morphologies of the brass substrates were comparatively observed by FE-SEM. For the untreated mesh, few bulges or crevices were observed on its smooth surface (A). The treated meshes contained numerous micro/nanoscale protrusions and pores (B and C). To identify the phases, composition, and crystallinity, the mesh with HP coating was further characterized by XRD (Appendix A) and XPS (Appendix A)**.** The characterized results for the untreated, electrodeposited and superhydrophobic brass meshes indicated that the HP was formed and a low surface energy material of FOTS was modified on the mesh surface after immersion.

Based on previous research [9,17], hydroxyapatite coatings with different surface wettability were obtained by altering the electrodeposition time. The optimal conditions were obtained with a constant current of 120 mA and the electrodeposited time of 60 min using the selected electrolytes with a pH of 5.6 in a water bath of 75 °C. As shown in Figure 2 (left), the largest WCAs and the lowest SA were obtained with an electrodeposited time of 60 min, indicating the successful fabrication of superhydrophobic surface. A series of ethanol-water droplets with different mass ratios were used for the determination of its surface wettability in Figure 2 (right). It is easily understood that a liquid droplet with less surface tension possesses a smaller CA value. When the mass ratio of ethanol was increased to 40 wt% (γ_lv_ = 33.88 mN/m), the corresponding CA is about 120.3 ± 2°. It is indicated that the prepared superhydrophobic mesh repelled liquid droplets with a surface tension value larger than 33.88 mN/m. Herein, CA is much less influenced by the hydrophilicity of the covered mesh than SA. In general, CA is related to the surface wettability and the SA reflects the adhesion behavior of the solid surface to the liquid. In our work, it was probably attributed to that the roughness of mesh surface was easily changed by the electrodeposited time relatively, but the surface hydrophobicity was intimately related to the mesh surface energy, so it was less influenced after being immersed in FOTS solution. Oil/water separation was tested using the fabricated brass mesh [18]. The mixed solution of 25 mL blue-dyed water and 25 mL red-dyed carbon tetrachloride was separated easily with the penetrated oil on the bottom of triangular flask and the blocked water on the top of superhydrophobic brass mesh in Appendix A.

Electrochemical corrosion performance of the fabricated brass meshes were detected in 3.5% NaCl at room temperature by an electrochemical workstation based on a traditional three electrode system, including counter electrode (platinum wire), reference electrode (saturated calomel electrode) and working electrode (the samples with exposed surface of 4 cm^2^). The samples were kept in NaCl solution for 30 min to obtain a stable open circuit potential before the test. The potential dynamic polarization curve test was performed under the scan rate 1 mv s^−1^. The polarization curves of three kinds of brass meshes after being soaked in 3.5 wt% NaCl solution for 5 days are given in Figure 3. Generally, the higher corrosion potential (Ecorr) and lower corrosion current density (Icorr), the better corrosion resistance [7]. The corrosion potential (Ecorr) increases from −0.530 V, to −0.514 V and −0.349 V for the studied brass meshes in Figure 3. According to the Tafel extrapolation, we can get that the corrosion current densities (Icorr) of the electrodeposited mesh and superhydrophobic coating are 4.10 × 10^−5^ A/cm^2^ and 3.77 × 10^−6^ A/cm^2^, respectively. Compared with the untreated mesh (4.26 × 10^−5^ A/cm^2^), it is observed that the Icorr of the superhydrophobic mesh is decreased by one order of magnitude, indicating its excellent anti-corrosion property for the superhydrophobic mesh. Due to more active sites created on its surface, the electrodeposited mesh is more prone to corrosion than the untreated one.

Besides, the corrosion resistance efficiency (η) could be calculated by Equation (1):(1)$$\eta =\frac{I_{corr}-I_{corr}^{\prime }}{I_{corr}}$$
where the *I_corr_* and *I_corr_*′ correspond to the corrosion current of untreated electrodeposited and superhydrophobic brass meshes, respectively. The corrosion resistance efficiencies of were 3.76% and 91.2% for electrodeposited and superhydrophobic meshes, respectively. The results of potentiodynamic polarization curves suggested that the superhydrophobic coating possessed an excellent corrosion resistance ability. On the one hand, the HP layer acted as a physical barrier to prevent the permeation and attack of water molecules and chlorides in the corrosion medium due to the ion-exchange and electrostatic interaction. On the other hand, the superhydrophobic surface reduced the contact area between the corrosion medium and the brass mesh due to the trapped air.

Scaling on the surface of industrial equipment was a common phenomenon, which seriously affected production efficiency and increased production costs [19]. A superhydrophobic coating would be a good candidate for anti-scaling because of the difficulty of deposition and adherence of fouling, which is caused by the low surface free energy. Herein, the scaling solution was prepared by mixing 250 mL 14.20 g/L Ca(NO_3_)_2_.4H_2_O and 250 mL 10.08 g/L NaHCO_3_ in a 500 mL beaker. The beaker with the mixed solution was kept in a water bath at 60 °C for several hours. Calcium will crystallize from a static oversaturated solution and precipitate in the form of calcium carbonate. The untreated, electrodeposited and superhydrophobic brass meshes were immersed into the oversaturated solution for the investigation of their surface scaling. Here, the deposition test of CaCO_3_ scaling was employed to illustrate the anti-scaling performance by weighing the dried meshes. As shown in Figure 4, the amount of CaCO_3_ scaling on the surface of the untreated electrodeposited and superhydrophobic brass meshes increased with the immersion time. The bare brass mesh had a higher scaling tendency (Figure 4) and the scaling process could be divided into two stages. In the first early stage from 1 to 8 h, the amount of CaCO_3_ scaling increased from 0.060 to 0.177 mg/cm^2^, with an average scaling rate of 0.0167 mg/(cm^2^·h). In the second stage from 8 to 20 h, the scaling rate decreased to 0.006 mg/(cm^2^·h), which was probably caused by the reduction in nucleation sites of CaCO_3_ scaling on the brass mesh surface. In the whole stage of crystal growth from 1 to 20 h, the average scaling rate of CaCO_3_ on the Cu mesh was 0.0125 mg/(cm^2^·h). The electrodeposited mesh exhibited similar anti-scaling ability to the untreated brass mesh from 1 to 13 h. However, for the superhydrophobic mesh, the average scaling rate of CaCO_3_ in the first stage (1–8 h), second stage (8–20 h) and the whole stage was 0.004 mg/(cm^2^·h), 0 mg/(cm^2^·h), 0.0016 mg/(cm^2^·h), respectively, which was lower than that of the untreated brass mesh. The main reason was that the superhydrophobic mesh could reduce the nucleation sites and prevent the deposition of CaCO_3_ at the surface.

Water drop impact experiments were conducted using a high-speed camera of Revealer at the rate of 15,000 frames per second. The unperturbed radius of the water droplet (10 μL) is r = 1.34 mm, and the impact velocity (v) is 0.45 ms^−1^, corresponding to We = 3.72, where We = ρv_0_^2^ r_0_/γ is the Weber number, with ρ being the density and γ the surface tension of liquid. Impinging from 1 cm height does not leave a small satellite droplet on the surface, which is convenient for the record of the rebounding height [20]. According to Figure 5a, it can be seen that during the process, the liquid droplet impacted and left the superhydrophobic surface quickly through three processes of spreading, shrinking and rebounding, suggesting that the superhydrophobic mesh had very low adhesion to the liquid droplets [21]. For DI water, the impact process of the water droplet lasted about five times longer due to the large surface tension of water droplet. Generally, for the same superhydrophobic surface, the greater surface tension of liquid, the better anti-wetting performance (the larger CA), resulting in a bigger jumped height for the droplet. As the droplets of TTAB concentration increased in the range of 0–6 mM in Figure 5a–f, their surface tension values decreased, and their maximum rebounding height values gradually decreased from 0.29 cm to zero and the wetting behavior was changed from superhydrophobic to hydrophobic. Finally, when the concentration of TTAB reached its CMC, the maximum bouncing heights and CAs tended to be stable.

High surface tension between a liquid droplet and a solid surface, results in a high CA and vice versa. The addition of surfactants decreases the surface tension, thereby decreasing the CA. The plots of surfactant-concentration−CA were constructed to determine the surfactant’s CMC value [22,23,24,25]. Herein, a plot of measured heights of the first bounce and CAs against TTAB concentrations were constructed, respectively, as shown in Figure 6A,B. The CMC value was determined from the break point of the maximum bounding height/CA versus the changed concentration curve. The determined CMC value of TTAB using the maximum bounding height (2.32 mM) and the CA method (1.91 mM) were obtained, which were in good agreement with its theoretic value of 2.1 mM with a relative error of 10% and 9%, respectively [26,27,28]. The measurement error might be improved by the usage of more precise instrument and proficient operator. Therefore, the CMC of other surfactants might be qualitatively calculated by measuring the changes of maximum bounding heights or static CAs on the superhydrophobic substrate.

## 4. Conclusions

In summary, we have successfully fabricated a robust brass mesh with superhydrophobicity and high oleophobicity. It was carried out by the forming of hydroxyapatite coatings through a constant current electro-deposition process, followed by immersion in fluoroalkylsilane solution. The resulted brass mesh with HP coating not only exhibited good prospects in anti-corrosion and anti-scaling fields, but also performed well in oil/water separation, the CMC determination of TTAB based on the Cas, and the maximum rebounding height of liquid droplets on the as-prepared superhydrophobic brass mesh. The CMC results obtained were close to the theoretic values of TTAB, indicating its potential to determine the CMC of other surfactants.

## Data Availability

Not applicable.

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
