# Peer review of "Robust Superhydrophobic Brass Mesh with Electrodeposited Hydroxyapatite Coating for Versatile Applications"

_molecules, 2022, doi:10.3390/molecules27175624_

Round 1

Reviewer 1 Report

The paper deal with the modification of brass mesh by the electro-deposition of hydroxyapatite followed by the reaction with perfluorooctyltriethoxysilane for the hydrophobization of the material and to bring anti-corrosion and anti-scaling properties.

Some aspects can be improved.

The role of hydroxyapatite (HA) coating can be better evidenced in the introduction part. HA was used to bring hydroxyl groups, to modify the roughness of the wires? The brass mesh can be directly hydrophobized with perfluorooctyltriethoxysilane?

Line 55 – a bibliography should be introduced.

The CMC was usually expressed in mM, mmol/L, mol/L, mmol∙L-1 but not mmol-1.

Please prove the doi or the web address of the reference 26. The CMC of TTAB surfactant in water was also determined in the literature using other methods (https://link.springer.com/article/10.1007/s10953-015-0318-0#Tab2        https://link.springer.com/article/10.1007/s003960000343 ). These articles can be also mentioned in the paper.

Author Response

1.The role of hydroxyapatite (HA) coating can be better evidenced in the introduction part. HA was used to bring hydroxyl groups, to modify the roughness of the wires? The brass mesh can be directly hydrophobized with perfluorooctyltriethoxysilane?

Answer: Thanks for the good suggestion! We have added some sentences about the description of hydroxyapatite (HA) in the introduction part. Though plasma spraying technique is the only clinically accepted method to deposit HA, it has several shortages including the usage of extremely high processing temperatures, the creation of a non-homogenous coating, and poor adhesion due to the delamination of HA from the substrate. During electro-deposition, the form of hydroxyapatite coating can lead to a rough surface for brass mesh with larger specific surface area. If the brass mesh was only hydrophobized with perfluorooctyltriethoxysilane directly, it is impossible for the brass mesh behaving the superhydrophobicity and high oleophobicity. What’s more, hydroxyapatite (Ca10(PO4)6(OH)2, HA) was generally used to improve the bioactivity and biocompatibility of titanium, magnesium, stainless steel and some other alloys, and thus is suitable for versatile application possibly.

  1. Line 55 – a bibliography should be introduced.

Answer: We cited one paper herein.

  1. The CMC was usually expressed in mM, mmol/L, mol/L, mmol∙L-1but not mmol-1.

Answer: The units of CMC have been corrected from mmol-1 to mM.

  1. Please prove the doi or the web address of the reference 26. The CMC of TTAB surfactant in water was also determined in the literature using other methods (https://link.springer.com/article/10.1007/s10953-015-0318-0#Tab2        https://link.springer.com/article/10.1007/s003960000343 ). These articles can be also mentioned in the paper.

Answer: Thanks for the kind suggestion, we have cited both papers in our revised manuscript. Reference 26 is a Chinese journal, there is only an article number of 1002-4956 (2013) 01-0044-02. And the web address is http://172.26.0.224/kcms/detail/detail.aspx?recid=&FileName=SYJL201301017&DbName=CJFDLAST2013&DbCode=CJFQ&uid=alhQNm9aSlZtWFd1bnBRcnY1UnpXbUcrUDF4QW50dURYRk04MlpxTTJ3VTd4engw

Reviewer 2 Report

The paper reports on a brass mesh covered with electrodeposited hydroxyapatite and made superhydrophobic by anchoring fluoroalkyl silane. Contact angles and critical micelle concentrations were determined.
Thus, the paper is in wide parts informative and well written. However, it still can and should be improved, thus the authors should consider my points mentioned below in a revised version of the paper.

Questions and remarks:

1.       Scheme 1: The term “magneton” is misleading; better use “magnetic stirrer”.

2.       Chapter 2.1: Is it really necessary to use a Pt mesh of the same size as the brass mesh as counter electrode? A smaller and cheaper Pt wire should work as well.

3.       Fig. 2a: It would be helpful for the reader, if you can shortly explain why the contact angle (CA) is much less influenced by the hydrophilicity of the covered mesh than the sliding angle (SA).

4.       Line 114: In chapter 2.1 it is said that current density, electrolyte temperature and deposition time were varied. In chapter 3 and Fig. 2a, however, you discuss only the time. Please give also the optimum other conditions.
By the way, why was the electrolyte concentration not varied – this should also be discussed shortly.

5.       Line 124: It must be “Fig. S3”.
Moreover, in the Supporting Data, the numbering of the Figures is not correct; Fig. 3 must be Fig. S1 and Fig. 4 must be Fig. S2.

6.       Line 129: It is obvious but should nevertheless be said, that the NaCl solution is aqueous.

7.       Line 142: better say “…one order of magnitude…” or “ – a factor of 10…”.

8.       Fig. 3: It is clear that the superhydrophobic brass mesh behaves best, but you should also comment shortly why the electrodeposited mesh is more prone to corrosion than the untreated one.

9.       Fig. 4: Why is curve B indicating that “The electrodeposited mesh didn’t exhibit the apparent anti-scaling ability” as said in line 177? That is not clear to me – please explain how you come to this conclusion.
Moreover, if for the superhydrophobic mesh the scaling rate is 0.004 in the first 8 h and about 0 in the next 12 h, the average over the whole time is lower than 0.004. Please correct this.

10.   Line 215: Can you shortly explain, why the CMCs determined from the maximum bounding height and the CA method differ by about 18%? Is that a general trend? Have also other surfactants than TTAB be tested?

11.   Conclusions, Line 226: Do you really mean “oleophobicity”? Why should the mesh be superhydrophobic and oleophobic at the same time, that is not probable?

Author Response

  1. Scheme 1: The term “magneton” is misleading; better use “magnetic stirrer”.

Answer: We have corrected magneton to “magnetic stirrer” in Scheme 1 in order to avoid misunderstanding.

  1. Chapter 2.1: Is it really necessary to use a Pt mesh of the same size as the brass mesh as counter electrode? A smaller and cheaper Pt wire should work as well.

Answer: It is a helpful suggestion, we will try it in our next work.

  1. Figure 2a: It would be helpful for the reader, if you can shortly explain why the contact angle (CA) is much less influenced by the hydrophilicity of the covered mesh than the sliding angle (SA).

Answer: Very good suggestion ! Herein, CA is much less influenced by the hydrophilicity of the covered mesh than SA. In general, CA is related to the surface wettability and the SA reflects the adhesion behavior of the solid surface to the liquid. The roughness of mesh surface is easily changed by the electrodeposited time relatively, but the surface hydrophobicity is less influenced after it is immersed in FOTS solution.

  1. Line 114: In chapter 2.1 it is said that current density, electrolyte temperature and deposition time were varied. In chapter 3 and Fig. 2a, however, you discuss only the time. Please give also the optimum other conditions. By the way, why was the electrolyte concentration not varied – this should also be discussed shortly.

Answer: Actually, we have done some experiments by changing the current density, electrolyte temperature and deposition time, but data handling was not systematically carried out due to too many figures. So we just focus on the investigation about the effect of deposition time. We have changed the description as follows: Based on the previous researches [9,17], hydroxyapatite coatings with different surface wettability were obtained by altering the electrodeposition time. The optimal conditions were obtained with a constant current of 120 mA and the electrodeposited time of 60 min using the selected electrolytes with a pH of 5.6 in a water bath of 75 °C.

  1. Line 124: It must be “Fig. S3”.
    Moreover, in the Supporting Data, the numbering of the Figures is not correct; Fig. 3 must be Fig. S1 and Fig. 4 must be Fig. S2.

Answer: Fig.3S has been corrected to Fig.S3, also, in the supporting file, the numbering of the Figures has been changed to Fig.S1 and Fig.S2, respectively, in our revised manuscript.

  1. Line 129: It is obvious but should nevertheless be said, that the NaCl solution is aqueous.

Answer: Herein, we have corrected the sentence as follow: The samples were kept in NaCl solution for 30 min

  1. Line 142: better say “…one order of magnitude…” or “ – a factor of 10…”.

Answer: we have corrected “1 order of magnitude” to “one order of magnitude” now.

8.Fig. 3: It is clear that the superhydrophobic brass mesh behaves best, but you should also comment shortly why the electrodeposited mesh is more prone to corrosion than the untreated one.

Answer: The electrodeposited mesh has more action sites than the untreated one, so it is more prone to corrosion than the untreated one. We have commented it shortly in line 142-143.

  1. Fig. 4: Why is curve B indicating that “The electrodeposited mesh didn’t exhibit the apparent anti-scaling ability” as said in line 177? That is not clear to me – please explain how you come to this conclusion.
    Moreover, if for the superhydrophobic mesh the scaling rate is 0.004 in the first 8 h and about 0 in the next 12 h, the average over the whole time is lower than 0.004. Please correct this.

Answer: Many thanks for the kind suggestion. We have used another suitable sentence to explain the conclusion. As shown in Fig.4, the electrodeposited mesh exhibited similar anti-scaling ability to the untreated brass mesh from 1 to 13h. For the superhydrophobic mesh, the average scaling rate over the whole time is 0.0016.

  1. Line 215: Can you shortly explain, why the CMCs determined from the maximum bounding height and the CA method differ by about 18%? Is that a general trend? Have also other surfactants than TTAB be tested?

Answer: For any analytical method, there is the determination error, which results from many factors including instrument, manual operation and etc. Negative relative error (9%) and positive relative error (10%) were obtained based on the determination of maximum rebounding height and CAs, respectively. There are many published papers about the determination based on CA. Herein, the present new method based non rebounding height could be used as an alternative for the determination of CMC. We are testing some surfactants such as SDS, CTAB and etc, and we believe the errors between the theoretic value and determined value can be decreased if we add more parallel experiments.

  1. Conclusions, Line 226: Do you really mean “oleophobicity”? Why should the mesh be superhydrophobic and oleophobic at the same time, that is not probable?

Answer: we are sure that the fabricated brass mesh is both superhydrophobic and olephobic. A series of ethanol-water droplets with different mass ratios were used for the determination of its surface wettability in Fig.2 (right). It is easily understood that liquid droplet with less surface tension possessed smaller CA value. When the mass ratio of ethanol was increased to 40 wt.% (γlv = 33.88 mN/m), the corresponding CA is about 120.3±2°. It is indicated that the prepared superhydrophobic mesh repelled liquid droplets with a surface tension value larger than 33.88 mN/m.

Many thanks for your kind suggestion and comment one more !

Round 2

Reviewer 2 Report

The authors responded very good to my questions in the first report.
However, I have the impression that the answers given, e.g. for questions/remarks 3, 10 and 11, are also inresting and important for the future readers.
Thus, more of the given explanations should also be included in the text in the manuscript.